# Modulation of the North Atlantic Deoxygenation by The Slowdown of the Nutrient Stream

Filippos Tagklis[1], Takamitsu Ito[1], and Annalisa Bracco[1]

1. Earth and Atmospheric Sciences, Georgia Institute of Technology, Atlanta, Georgia, USA.

*Correspondence to*: Filippos Tagklis (ftagklis3@gatech.edu)

**Abstract.** Western boundary currents act as transport pathways for nutrient-rich waters from low to high latitudes (nutrient streams) and are responsible for maintaining mid- and high-latitude productivity in the North Atlantic and North Pacific. This study investigates the centennial oxygen ($O_2$) and nutrient changes over the Northern Hemisphere in the context of the projected warming and general weakening of the Atlantic Meridional Overturning Circulation (AMOC) in a subset of Earth System Models included in the CMIP5 catalogue. In all models examined, the Atlantic warms faster than the Pacific Ocean, resulting in a greater basin-scale solubility decrease. However, this thermodynamic tendency is compensated by changes in the biologically-driven $O_2$ consumption which dominates the overall $O_2$ budget. These changes are linked to the slow-down of the nutrient stream in this basin, in response to the AMOC weakening. The North Atlantic resists the warming-induced deoxygenation due to the weakened biological carbon export and remineralization, leading to higher $O_2$ levels. On the contrary, the projected nutrient stream and macro-nutrient inventory in the North Pacific remain nearly unchanged.

## Introduction

Deoxygenation of the oceans is potentially one of the most severe ecosystem stressors resulting from global warming given the high sensitivity of dissolved oxygen to ocean temperatures. Unrestrained anthropogenic $CO_2$ emissions and consequent warming are likely to disrupt marine habitats and influence the cycles of biogeochemically essential elements (Gruber, 2011). Global-scale deoxygenation has taken place during the second half of the 20th century (Stramma et al., 2008), and a widespread recognisable signal of $O_2$ decline is emerging beyond the envelope of natural variability (Schmidtko et al., 2017;Ito et al., 2017). The Earth Systems Models (EaSMs) included in the CMIP5 (Coupled Model Intercomparison

Project – Phase 5) catalog project a robust (across models) decline in dissolved $O_2$ inventory for the $21^{st}$ century despite differences in models' complexity, biogeochemical parameterizations and warming responses. Under the "business as usual" scenario, all models predict enhanced hypoxic conditions and dissolved oxygen loss (Bopp et al., 2013;Cocco et al., 2013).

The dissolved oxygen is controlled by air-sea exchange, circulation, and biology, and the dissolved oxygen concentrations in the interior ocean reflect a balance between ventilation, circulation and biological consumption. Warming climate can cause shifts in this balance. The solubility of dissolved oxygen is inversely proportional to seawater temperature, and air-sea $O_2$ exchange is a relatively fast process in the ice-free open ocean, of the order of $O$(20 days) (Broecker and Peng, 1974;Wanninkhof, 1992). All else unchanged, in a warming climate there would be a corresponding $O_2$ decline closely following the temperature-solubility relationship of seawater (Najjar and Keeling, 1997). However, changes in ocean stratification, ventilation and biological productivity can further change dissolved oxygen. During the transient trajectory of the climate system as it adjusts to anthropogenic forcing, near-surface waters warm faster than deeper waters, leading to an increase in ocean stratification. In a more stratified ocean, the ventilation of sub-surface waters diminishes, reducing the $O_2$ supply to the ocean interior (Bopp et al., 2002;Frölicher et al., 2009). Furthermore, increased stratification is expected to weaken the meridional overturning circulation and therefore the ventilation of the waters deeper than 1000 m (Meehl and Stocker (2007). At the same time, the weakening of the overturning circulation may decrease the overall vertical mixing and therefore the supply of nutrient-rich waters to the euphotic layer, thus causing a reduction in biological productivity and carbon export. As upwelling becomes less effective in uplifting nutrient-rich waters, export production of organic material and oxygen consumption through respiration also diminishes, but as water parcels spend more time in the ocean interior, the oxygen consumption integrated over time may increase (Rykaczewski and Dunne, 2010).

Western Boundary Currents (WBCs) plays an essential role in biogeochemical cycling. In the northern hemisphere, WBCs represent an advection pathway for nutrients from the ocean boundaries into the open waters. They are known as "nutrient streams" and are responsible for maintaining basin-scale high productivity in the mid- and high-latitudes over interannual and longer timescales (Letscher et al.,

2016;Palter et al., 2005;Williams et al., 2011;Williams et al., 2006). High nutrient concentrations extend from tropical coastal areas into the interior of the Pacific and Atlantic Oceans, following the Kuroshio Current and the Gulf Stream (Pelegrí and Csanady, 1991). From a dynamical perspective, recent studies have shown that the nutrient supply due to the lateral transport in the subtropical euphotic zone dominates over the vertical transport (Letscher et al., 2016), with mean and eddy horizontal cross-boundary nutrient transport accounting for ~75% of the total nutrient supply into the subtropical gyres (Yamamoto et al., 2018). Therefore, changes in this horizontal nutrient transport, through changes in the WBC characteristics, can have a profound influence on the basin-scale biogeochemical cycling.

The primary objective of this study is to investigate how and why the dissolved oxygen content of the North Atlantic and the North Pacific basins is projected to change in the 21st century using a suite of EaSM integrations. In particular, we aim at understanding and quantifying the role of the nutrient streams in the centennial scale deoxygenation and nutrient loading of these two basins. We first verify the EaSMs' skill in reproducing the mean state of relevant biogeochemical variables and then analyze the model projections to the end of the 21st century.

**Data and Methods**

For this study, we analyse seven CMIP5 EaSMs for which the variables of interest are available. The suite includes two versions of the Geophysical Fluid Dynamics Laboratory (GFDL) Earth System Model, GFDL-ESM2G and GFDL-EASM-2M (Dunne et al., 2013;Dunne et al., 2012), the Community Earth System Model, CESM1-BGC  (Long et al., 2013;Moore et al., 2013a, b), two versions of the Institute Pierre Simon Laplace model, IPSL-CM5A-LR and IPSL-CM5A-MR (Dufresne et al., 2013), and two of the Max Plank Institute model, MPI-ESM-LR and MPI-ESM-MR(Giorgetta et al., 2013a;Giorgetta et al., 2013b). The EaSMs vary regarding the parameterisations of the ocean circulation and biogeochemical modules, but the biogeochemical component in all cases is formulated as Nutrient-Phytoplankton-Zooplankton-Detritus (NPZD) type. For each member, we examine the last 30 years (1970-2000) of the twentieth century in the historical simulations and the last 30 years (2070-2100) of the

twenty-first century under the future projections based on the Representative Concentration Pathway 8.5 scenario or "rcp8.5" (Riahi et al., 2011a;Taylor et al., 2012;Riahi et al., 2011b).

All the variables used in the CMIP5 analysis are three-dimensional and annually averaged fields interpolated onto a common 1° x 1° longitude-latitude grid domain and 33 depth levels, consistent with the World Ocean Atlas. The interpolation method applied was bilinear using the Climate Data Operators (Schulzweida, 2019). The variables of interest are dissolved oxygen ($O_2$), temperature (T), phosphate ($PO_4$), particulate organic carbon export at 100m depth (EP) and current speed ($CS = V_{CS} = (\sqrt{u^2 + v^2})$) in units of meters per second. Oxygen solubility ($O_{2,sat}$) is calculated from potential temperature and salinity following Garcia and Gordon (1992). Apparent oxygen utilisation (AOU) is then determined as the difference between the $O_{2,sat}$ and $O_2$ (AOU= $O_{2,sat}$ - $O_2$). AOU changes quantify contributions from processes other than warming, such as remineralisation of organic matter and/or the rate of transport and mixing of water mass (Sarmiento and Gruber, 2006). The separation of oxygen changes $\Delta O_2$ into a biologically/transport-driven component, $\Delta(AOU)$, and a thermodynamically-driven component, $\Delta O_{2,sat}$, is based on the assumption that the surface oxygen is always in equilibrium with the overlying atmosphere. However, intense air-sea interactions during wintertime at the high latitudes often cause under-saturated surface $O_2$, leading to a non-negligible preformed AOU (Ito et al., 2004). Unfortunately, stored variables in the model outputs do not allow a more precise estimation.

It has been shown that in the CMIP5-EaSMs the biogeochemical tracers are not always equilibrated with respect to the ocean circulation. To account for the magnitude and sign of this model drift, in all analyses we used the pre-Industrial Control simulations (piControl) and removed the drift by defining, for example, $O_{2_{trend}} = \left\{ O_2^{rcp8.5(B)} - O_2^{hist(A)} \right\} - \left\{ O_2^{piControl\ (B)} - O_2^{piControl\ (A)} \right\}$ where A and B indicate the periods 1970-2000 and 2070-2100.

**Results**

**Model Evaluation**

We first evaluate the model representation of the distributions of key biogeochemical variables, including $PO_4$, $O_2$ and AOU. We focus on the Northen Hemisphere (10°N-65°N) and concentrate on the upper layer

of the ocean (depth range 0-700 m). The CMIP5 climatological values are calculated over the period 1970-2000 in the "esmHistorical" experiments. Annual mean climatologies from the World Ocean Atlas 2009 (WOA09) (Locarnini et al., 2010;Garcia et al., 2010;Antonov et al., 2010)  are used as an observational reference. Note that in Figures 1-3 the Pacific and Atlantic basins are plotted in separate panels with different color scales because of the large differences in their mean values.

The observed $PO_4$ concentrations **[Figure 1]** range from ~0.8μM in the subtropical North Pacific (STNP) gyre to values greater than 2.7μM in the subpolar North Pacific (SPNP) gyre and the eastern boundary and equatorial upwelling region at the lower latitudes. The EaSM are broadly in agreement over the North Pacific regarding the $PO_4$ spatial gradients, with the exception of  CESM1-BGC  that underestimates the latitudinal differences with higher nutrient levels overall. In all models, there is a slight underestimation of  $PO_4$ in the subpolar region, that is reflected in the multi-model mean (MMM) where values are about ~0.3μM smaller than in the WOA09. In the North Atlantic, the observed concentrations range from ~ 0.2μM in the subtropical (STNA) gyre to ~1.15μM in the subpolar (SPNA) gyre. In contrast to the Pacific ocean, there are significant model-to-model differences in the $PO_4$ spatial pattern. All models but IPSL-CM5A-LR overestimate the concentrations of $PO_4,$ with CESM1-BGC displaying the largest bias, followed by GFDL-ESM2M.

The simulated pattern of dissolved oxygen is better captured than $PO_4$ by each model individually and therefore by the MMM, especially in the Atlantic basin. In the Pacific ocean, the observed dissolved oxygen concentrations range from ~160μM in the STNP gyre to ~50μM in the SPNP gyre. GFDL-ESM2M, IPSL-CM5A-LR and IPSL-CM5A-MR overestimate dissolved oxygen in the STNP by ~35μM and CESM1-BGC and MPI-ESM-MR underestimate oxygen concentration in the same area. The end result is a MMM that compares relatively well to WOA09 due to the compensating biases. In the North Atlantic the concentrations of dissolved oxygen range from ~180μM in the STNA gyre to ~340μM in the western SPNA and ventilation sites. The latitudinal gradient reflects both the temperature gradient and the presence of well-mixed and ventilated cold subpolar waters.

In terms of AOU, the CMIP5-ESMs integrations capture the observed climatological distribution with more robust (across models)  patterns in the Atlantic region [Figure 3]. In the Pacific Ocean, the

AOU concentrations range from ~30µM in the STNP gyre to ~250µM in the SPNP gyre. The overall higher values of AOU in the Pacific compared to the Atlantic basin are due to the older age of the waters and the limited physical $O_2$ supply to intermediate and deep waters. In the Atlantic Ocean, low AOU values are found in the SPNA as convection and deep water formation decrease the AOU in this region. The narrow band of higher AOU values around ~60µM that extends from the tropics to the east into the basin following the Gulf Stream and the North Atlantic Current (NAC) pathway is captured by all models with different intensity, and is present in the MMM, even if slightly weaker than observed due to biases in the representation of the Gulf Stream separation and NAC location. The skill of each model in capturing the mean nutrient concentration **[Figure 1]** is also reflected in the intensity of AOU **[Figure 3].** For example, CESM1-BGC as the one extreme in the North Atlantic, overestimating (>1.2µM) the nutrient concentration, shows the highest (>70µM) AOU values, while both IPSL versions underestimate the nutrient concentrations and show low AOU values (<70µM).

**Centennial Changes**

We next examine hemispheric centennial changes of the physical and biogeochemical variables in the North Pacific and North Atlantic oceans. Changes are calculated as the differences between the 30-year period 2070-2100 in the rcp8.5 scenario and 1970-2000 in the historical simulations. We use 30-year periods to ensure that year to year changes are mostly averaged out. For $O_2$, T and AOU we also verify the statistical significance of the drift-corrected trends by testing if the average concentrations during 2070-2100 under the rcp8.5 scenario are significantly lower than those during 1970-2000 period relative to the interannual variability within each 30-year period. We did so using a t-test and evaluating $t = \dfrac{-\{(\overline{x_{rcp8.5}} - \overline{x_{his}}) - \Delta x_{piControl}\}}{\sigma \sqrt{\frac{1}{N_1} + \frac{1}{N_2}}}$ where σ is defined as $\sqrt{\dfrac{N_1 s_1^2 + N_2 s_2^2}{N_1 + N_2 - 2}}$, and the degree of freedom is d.f.=$N_1$+$N_2$-2. In our case, the number of records in each sample set is the same N=$N_1$=$N_2$=30 and $s_1$, $s_2$ the corresponding sample variance. Preindustrial control simulations are used to correct for the model drift as mentioned earlier.

Under the rcp8.5 scenario, both basins warm by 0.5 - 4°C [**Figure 4]**, and the warming is generally stronger in the Atlantic than in the Pacific. A localised patch of cooling stands out in the SPNA in all

models but in different locations. This patch is known as "warming hole" (Drijfhout et al., 2012;Rahmstorf et al., 2015a, b) and is a response to the reduced poleward transport of heat due to the AMOC slowdown, which is common to all models (Tagklis et al., 2017). The location of the warming hole depends on each model representation of the NAC pathway. Despite the presence of this cold patch, basin-scale averages between $10^{o}N$-$48^{o}N$, shown in Table 1, reveal that the North Atlantic takes up more heat than the Pacific, and warms on average $\Delta T \sim 1^{o}C$ more than the Pacific. This mean difference is consistent across the models. Additionally, in the Atlantic the warming pattern is consistent among the models, with stronger warming at the gyre boundaries, both at the tropical-subtropical and subtropical-subpolar boundaries.

Even though the Atlantic ocean is warming faster than the Pacific, the centennial changes of $O_2$ in Figure 5 reveal a more moderate deoxygenation rate in the Atlantic compared to the Pacific. The trends shown in the figure are statistically significant nearly everywhere, according to a t-test at the 99% confidence level. The oxygen trend in the Atlantic is "patchy" with the subtropics resisting to deoxygenation especially in correspondence of the Gulf Stream/NAC paths (Tagklis et al., 2017). The subpolar regions offshore Newfoundland and Labrador, on the other hand, lose the most oxygen in this basin, in correspondence with the largest warming signal. The basin scale averages in Table 1 confirm that the Atlantic Ocean is losing oxygen at a lower rate than the Pacific in all seven models. The modelled basin averaged $O_2$ changes are between -3.4 and -12 µM but mostly in the -6 µM range in the Atlantic and between -10 to -18.1 µM in the Pacific. This corresponds to $O_2$ decline of 3% in the Atlantic and 10% in the Pacific compared to their 1970-2000 mean state.

The inverse proportionality of the solubility of oxygen to seawater temperature implies that negative/positive changes in temperature are reflected as positive/negative changes in oxygen solubility $\Delta O_{2,sat}$ . In thermocline waters, a temperature increase by 1°C causes a solubility decrease of about 7µM. Given the modeled warming trends, oxygen solubility decreases in both basins for all seven models, except for the warming holes in the SPNA. The rate of solubility change in the Atlantic Ocean ranges from -8.4 µM for GFDL-ESM2G to -12.2 µM for IPSL-CM5A-MR; in the Pacific Ocean ranges from -

5.1 µM for MPI-ESM-LR to -8.6 µM for IPSL-CM5A-LR (see Table 1). The solubility decline is more pronounced in the subpolar Atlantic as expected, but this is in contrast to the net $O_2$ change in all models.

The AOU signal explains the different $O_2$ trend **[Figure 6]**. In the subtropical regions, the AOU decreases in all models in the North Atlantic, but increases overall in the Pacific, even if with inter-model regional differences. As the ocean's surface warms and becomes more stratified, AOU generally increases

due to weakened ventilation and sustained biological $O_2$ consumption which dominates over the physical supply. The effect of respiration is accumulated as water spends more time in the ocean interior, leading to a decline of $O_2$. This is verified in most of the North Pacific and in the subpolar North Atlantic. In the subtropical North Atlantic, however, AOU and stratification decouple due to changes in lateral transport and biological oxygen utilization as shown next. Basin-scale averages of $\Delta$AOU in Table 1 are in the

range –1.5 µM for GFDL-ESM2G to -6.6 µM for MPI-ESM-2M in the Atlantic and in the range +4.6 µM for MPI-ESM-LR to +10.65 µM for CESM1-BGC in the Pacific. The question that naturally follows is: how could the subtropical North Atlantic have a significant decrease in AOU under the increasing stratification? It is unlikely that the thermocline ventilation increases under this condition. Also, the mechanism at work must be specific to the North Atlantic Ocean.

In all EaSMs examined the speed of the Gulf Stream and NAC extension decreases; in contrast, the speed of the Kuroshio Current does not change noticeably **[Figure 7]**. Consequently, the "nutrient stream" in the North Atlantic loses part of its strength. Since it is a major supply pathway of macro-nutrients for the North Atlantic, the nutrient inventory and the biological productivity decline in the subtropical gyre. This mechanism is confirmed by the significant decline of the $PO_4$ inventory projected

in the North Atlantic **[Figure 8]**, and by the weakening in carbon export in all models **[Figure 9]**.

The weakened remineralization results in the regional AOU decline, which can compete against the effect of weakened ventilation. In the North Pacific, on the other hand, the $PO_4$ inventory displays a moderate increase, again following the currents' behaviour. Basin-scale averages of $\Delta PO_4$ in Table 1**,** range from -0.006 µM for IPSL-CM5A-LR to -0.16 µM for CESM1-BG. In the North Pacific, the nutrient

decline is close to zero. Further support for this proposed mechanism can be found in the North Atlantic in the IPSL-CM5A-LR model where the weakest current speed decline **[Figure 7]** is associated with the

weakest PO$_4$ decline (-0.006 μM), the strongest warming and stronger deoxygenation (-12 μM; Table 1) among the models.

It is important to note there is no overall agreement in the patterns or signs of centennial changes in export production, $\Delta EPC_{100}$, among the models. Also, the pattern of the carbon (C) export does not necessarily correspond to the changes in AOU, which instead follow the concomitant changes in ventilation. AOU reflects the integrated respiration rates over the ventilation pathways, so it is not surprising that the patterns look different between $\Delta AOU$ and $\Delta EPC_{100}$. Having said this, it is expected that basin-scale decrese in carbon export and respiration are likely to cause a decrease in AOU. The C export decreases globally, but the magnitude of the decline is particularly strong in the North Atlantic **[Figure 9]**. It generally decreases under increasing stratification because of the reduced upwelling and entrainment of subsurface macro-nutrients, which partially compensates the deoxygenation due to the reduced ventilation. The net effect on the AOU is dominated by ventilation in the North Pacific and the subpolar North Atlantic. However, this is not the case in the subtropical North Atlantic. The decline of the C export is much stronger due to the compounding impacts of the increased stratification and the weakened North Atlantic nutrient stream, as evidenced by the decline in the phosphorus inventory **[Figure 8]**. This is consistent with the decline in nutrient supply in the North Atlantic and the resultant decrease in AOU. On the contrary, the AOU in the subtropical gyre of the North Pacific increases, despite the weakened C export, suggesting that the weakened ventilation in this region contributes the most to deoxygenation.

In Figure 10 and Figure 11, we further analyze the mechanisms at play in the Atlantic Basin. To investigate more in depth the nutrient inventory changes in the North Atlantic, we estimate changes of the northward supply of phosphate at 10°N along with the nutrient inventory of the subtropical gyre. Figure 10 time series represent the zonally and vertically integrated northward transport of phosphate ($\overline{vPO_4}$) at 10°N over the 0-700 meters depth range, decomposed in the overturning ($MO = \bar{v}\overline{PO_4}$) and gyre ($GY = \overline{v'PO'_4}$) components, along with the nutrient inventory (NI) zonally, meridionally and depth-integrated over 10°N-48°N and 0-700 meters. The overbar indicates the zonal mean, and the primes indicate the departure from the zonal mean. For better comparison, we apply a low pass filter of 10-years,

and then we normalise the time series by subtracting their mean and divide by their standard deviation. The coloured values represent the percent centennial change of each transport component and nutrient inventory. The subtropical gyre nutrient inventory closely follows the declining trajectory of the overturning component of the northward nutrient transport at 10°N for all models but IPSL-CM5A-LR.

To better understand the reduction in carbon export, we explore the nutrient supply to the surface euphotic layer through the vertical entrainment of thermocline nutrients. If averaged over a broad area, the downward export of organic matter is mostly replenished by the upwelling and vertical mixing of nutrients during cool seasons. Over the subtropical oceans, wind-driven Ekman downwelling dominates the mean large-scale circulation, so the winter-time deepening of the mixed layer is the primary pathway for the vertical nutrient supply. The entrainment flux of nutrient (P) can be represented as $E_{flux} = \wedge (P_{th} - P_m)\frac{\partial h}{\partial t}$, with the operator $\wedge = 1$ when the mixed layer thickness increases $\frac{\partial h}{\partial t} > 0$. $(P_{th} - P_m)$ is the vertical difference in nutrient concentration between the thermocline and mixed layer. Integrating over one year, we approximate the annual entrainment as $E_{flux,ann} \sim H * dPz$, where H is the difference between the yearly maximum and minimum mixed layer depth, and $dPz$ is the vertical nutrient difference between the surface and 300m. The centennial changes of those two terms are presented in Figure 10 as percent changes along with the changes of the $\Delta EPC_{100}$. The change in sign of both terms $\Delta H$ and $\Delta(dPz)$ across SPNA and STNA are in response to different processes. We direct the reader's attention to the lower panels of Figure 10 and the multi-model mean behavior. In the subpolar regions, the maximum mixed layer depth is significantly reduced ($\Delta H < 0$) with the suppression of convective mixing, while the vertical gradient of the nutrient increases($\Delta dPz > 0$). This indicates that the reduction in export production is primarily caused by the increased stratification and weakened vertical mixing of nutrients into the surface euphotic layer. The increased vertical nutrient gradient cannot cause the weakened export production. On the contrary, in the subtropical Atlantic region, the maximum mixed layer depth deepens with time ($\Delta H > 0$) along the WBC. Positive $\Delta H > 0$ tends to increase the entrainment of nutrient in the euphotic layer but the vertical gradient of the nutrient decreases ($\Delta dPz < 0$), as a result of the total nutrient inventory decline. The reduction in export production in the subtropics is likely caused by the weakened vertical gradient of nutrient. Increased seasonality of the mixed layer depth cannot explain the reduction

in export production. The close relationship between the reduction in basin-scale nutrient inventory and the zonal mean (meridional overturning) nutrient transport indicates that the weakened nutrient stream is causing the weakened export production in the subtropical North Atlantic.

Finally, in Figure 11, we include an estimate of the regional nutrient budget using GFDL-ESM2M as a representative model. Following the geographical boundaries of the previous analysis, we consider a control volume enclosing the STNA with boundaries at approximately $10^{\circ}N$-$48^{\circ}N$, $80^{\circ}W$-$10^{\circ}W$ and 700 meters depth using the native model grid. We calculate all lateral nutrient transport terms in and out of the control volume, in units of moles per second. Zonal and meridional fluxes are defined as positive eastward and northward, and the veritical flux is defined as positive upward.

Figure 11a shows the changes in magnitude and the sign of each component during the historical period 1861-2005 and rcp8.5 period 2006-2100. The northward supply of nutrients at $10^{\circ}N$ ($vPO4_{(10oN)}$) has the most significant decline among the lateral transport terms and magnitude comparable to the northward transport of nutrient at $48^{\circ}N$ ($vPO4_{(48oN)}$). The western boundary transport component at $80^{\circ}W$ ($uPO4_{(80^{\circ}W)}$) represents the net nutrient supply through the Florida current which decreases to half of its magnitude by the end of the 21$^{st}$ century. The eastern boundary component at $10^{\circ}W$ ($uPO4_{(10^{\circ}W)}$) remains largely unchanged. The vertical component at 700 meters depth ($wPO4_{(700m)}$) is negative (downwelling) with decreasing magnitude. The signs and the magnitude of the changes in the lateral and vertical transport terms are consistent with a weakening of the advective nutrient transport, providing additional support to our interpretation.

The flux convergence of the (resolved) advective transport must be balanced by the time derivative of the nutrient inventory and the net biological nutrient sources/sinks. Sub-grid scale parameterizations could also contribute to the regional nutrient budget. It is difficult to precisely close the nutrient budget with available dataset. However, we can still integrate over time the flux convergence of the advective transport to calculate the 'estimated' nutrient inventory ($PO4_{estimated}$). Net advective convergence is positive, and its integral gradually increases over time because it does not include the nutrient uptake and export by biological processes (Bio). To account for the baseline pre-industrial biological component (Bio), we first determine the residual between the $PO4_{estimated}$ and the PO4 as Residual ($=PO4_{estimated}$ -

PO4). The pre-industrial estimate of Bio is then estimated as a linear trend based on the first 60 years (1861-1920) of the Residual and the corrected PO4$_{estimated}$ is determined as the temporal integral of the advective flux convergence minus Bio. The PO4$_{estimated}$ time series reflects the change in STNA nutrient budget if there were no changes in biological sources/sinks (constant Bio). Figure 11b shows that the decline of PO4$_{estimated}$ is much larger than that of PO4. After 2005 and during the rcp8.5 period the actual PO4 inventory does not decrease as much as the estimated inventory PO4$_{estimated}$ due to the weakened biological export of nutrients. This result is consistent with a weakening of the biological productivity of the North Atlantic as well as the circulation change as drivers of the nutrient decline in the basin.

## Conclusions

We analyzed a subset of seven EaSMs included in the CMIP5 catalogue to understand current and future state of oxygen distribution in the upper 700 m of the water column in the northern hemisphere. During the historical period 1970-2000, models reproduce the observed mean state of dissolved oxygen concentration, capturing spatial variations in apparent oxygen utilisation and, most importantly, reproduce the "nutrient stream". By the end of this century, the upper water column in the business as usual scenario is projected to warm more in the North Atlantic compared to the North Pacific. Despite this tendency, the subtropical North Atlantic resists to deoxygenation. As the ocean warms, $O_2$ saturation decreases globally, with the exception of the warming holes in the North Atlantic, but the two basins differ especially in the AOU. In the subtropical North Atlantic, the basin-mean AOU decreases and is decoupled from the stratification-induced reduction in ventilation. In all models but one (IPSL-CM5A-LR), the AMOC weakening is associated with a decline in the current speed of the Gulf Stream and its extension and, in turn, to a decline in the nutrient stream. Lateral nutrient supply, quantified by the reduction in phosphate inventory, decreases, and so does biological productivity, as confirmed by the negative trend in ΔEPC100. The decline in biological productivity and consequent retention of $O_2$ (by weakened biological consumption) in the subtropical North Atlantic are sizable enough to compensate the $O_2$ solubility trend. The decline in the nutrient stream is not detected in the North Pacific, where biological productivity does not change as dramatically as in the Atlantic, and the solubility trend dominates.

Our results imply that the ocean deoxygenation progresses more intensely in the North Pacific Ocean even though its heat uptake is moderate compared to its neighbour ocean. This faster and stronger decline appears to be supported by the relatively stable P inventory of the North Pacific. The macronutrient inventory of the North Pacific is "charged up" with the higher concentrations of nutrients in comparison to the North Atlantic due to the old age of the Pacific waters. In contrast, the North Atlantic nutrient inventory is more dynamic given that nutrient streams critically depend on the AMOC and its feedbacks on the western boundary current system. This difference has significant consequences given that the background, climatological $O_2$ levels are much lower in the Pacific basin, again due to the older age. The Pacific Ocean indeed hosts already two of the four most voluminous oxygen minimum zones. Higher rate of $O_2$ loss can potentially lead to more frequent and intense hypoxic events, with devastating impacts for the marine ecosystem (Penn et al., 2018). The length of the EaSM integrations does not allow to verify if the reduction in the biological activity of the subtropical North Atlantic is only transient, and if a rebound may take place once a new climate equilibrium is achieved (Moore et al., 2018). Further investigations and higher resolution model outputs are also needed to better constrain the regional patterns of biological productivity and oxygen changes.

**Acknowledgments**

We acknowledge the World Climate Research Programme's Working Group on Coupled Modeling, which is responsible for CMIP, and the U.S. Department of Energy's Program for Climate Model Diagnosis and Intercomparison for providing support and software infrastructure in partnership with the Global Organization for Earth System Science Portals. The CMIP data were obtained from the CMIP5 Data Access Portal (http://cmip-pcmdi.llnl.gov/cmip5/data_portal.html). The Word Ocean Atlas Data (WOA09) were obtained from the National Center for Environmental Information (www.nodc.noaa.gov/OC5/SELECT/woaselect/woaselect.html). We thank two reviewers and the Editor for their useful comments.

We acknowledge the support by the NOAA Climate Program Office, Climate Variability and Predictability Program, through grant NA16OAR4310173. In its initial stages this work was supported by a grant from the National Science Foundation (NSF-OCE 097394084).

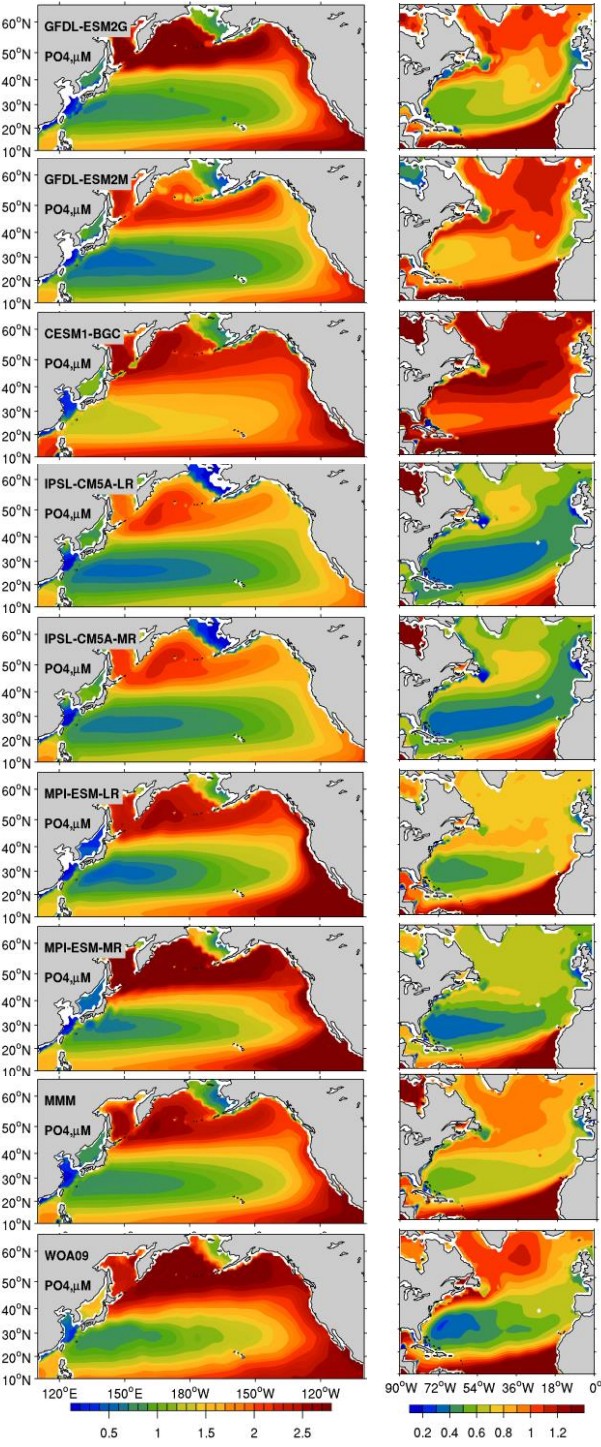

**Figure 1 : Upper ocean (0-700 m) concentration of phosphate (PO4), for the period 1970-2000 in a subset of the CMIP5 models (esmHistorical), Multi-Model-Mean (MMM), and World Ocean Atlas 2009 (WOA09). The North Pacific and Atlantic basins are plotted with different colour ranges to better highlight the spatial patterns in models and observations.**

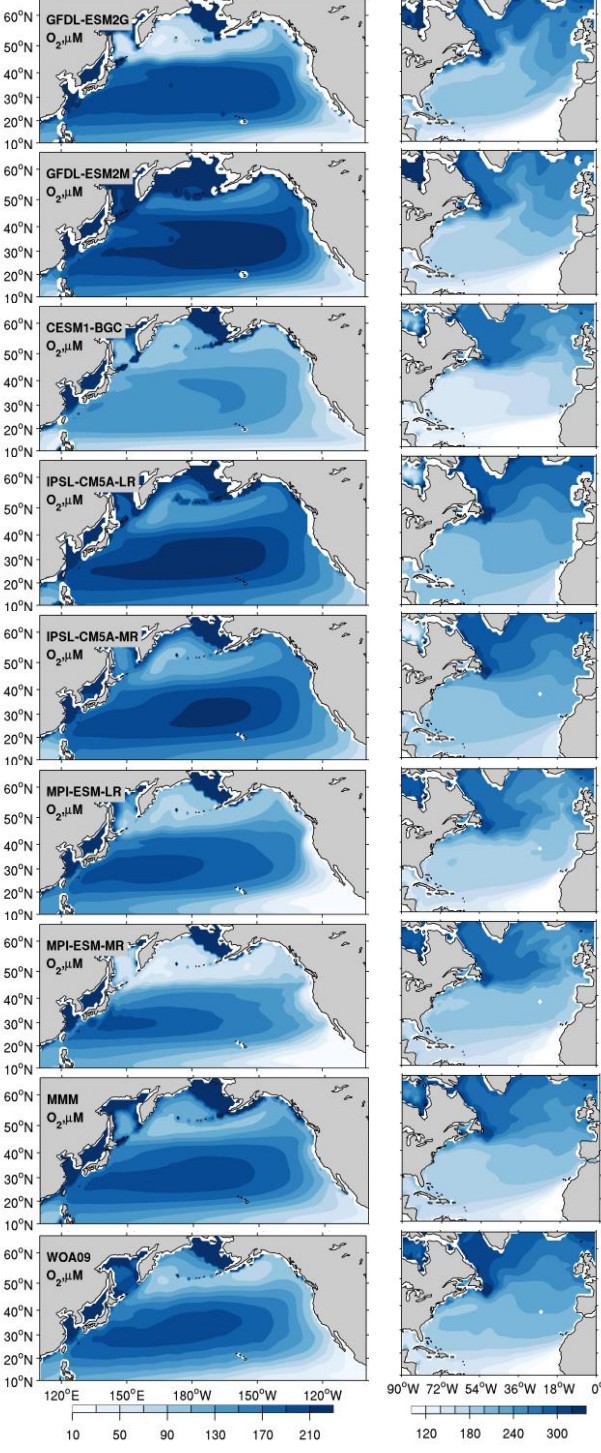

**Figure 2 Upper ocean (0-700 m) concentration of dissolved oxygen (O2), for the period 1970-2000 in a subset of the CMIP5 models (esmHistorical), Multi-Model-Mean (MMM), and World Ocean Atlas 2009 (WOA09). The North Pacific and Atlantic basins are plotted with different colour ranges to better highlight the spatial patterns in models and observations.**

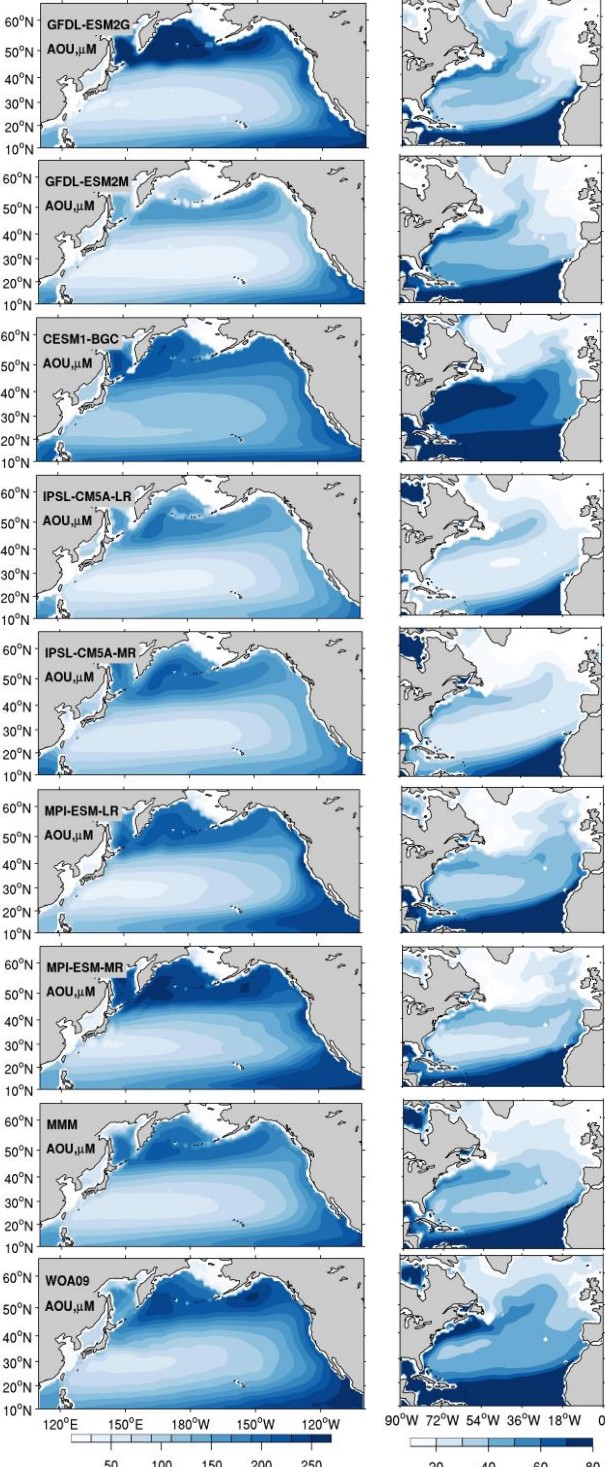

**Figure 3 Upper ocean (0-700 m) concentration of apparent oxygen utilization (AOU), for the period 1970-2000 in a subset of the CMIP5 models (esmHistorical), Multi-Model-Mean (MMM), and World Ocean Atlas 2009 (WOA09). The North Pacific and Atlantic basins are plotted with different colour ranges to better highlight the spatial patterns in models and observations.**

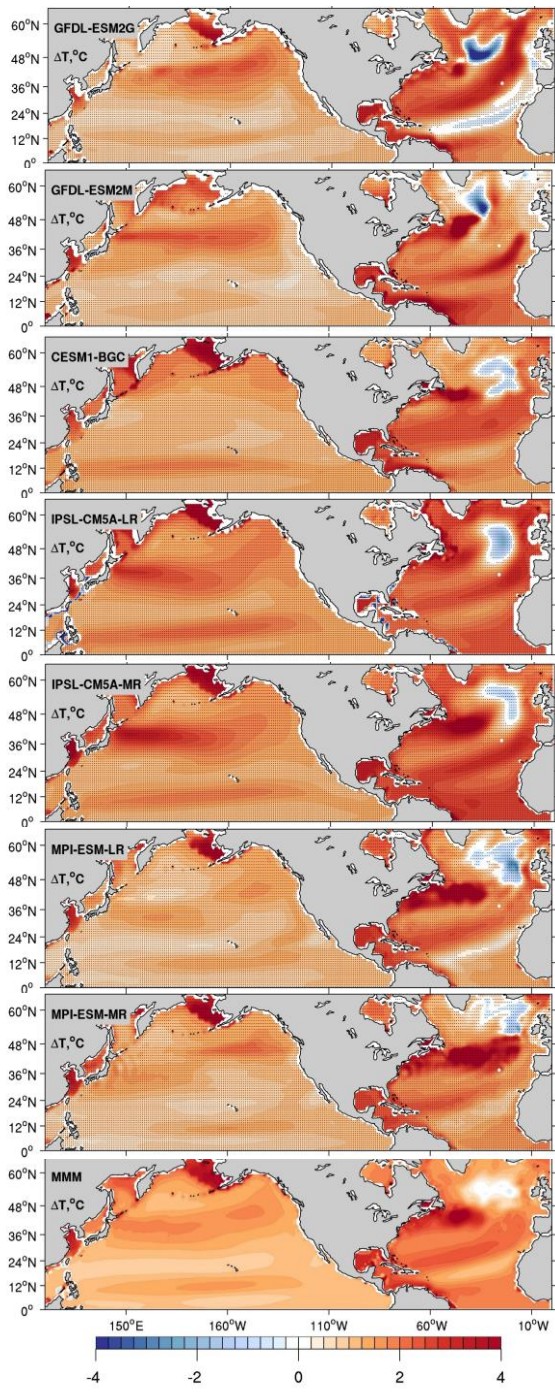

**Figure 4** Centennial change of T calculated as the difference in 30-year averages between (2070-2100) and (1970-2000). All plotted values are 0-700 m depth averages. Black dots indicate areas where the results are statistically significant at the 99% confidence level according to a t-test.

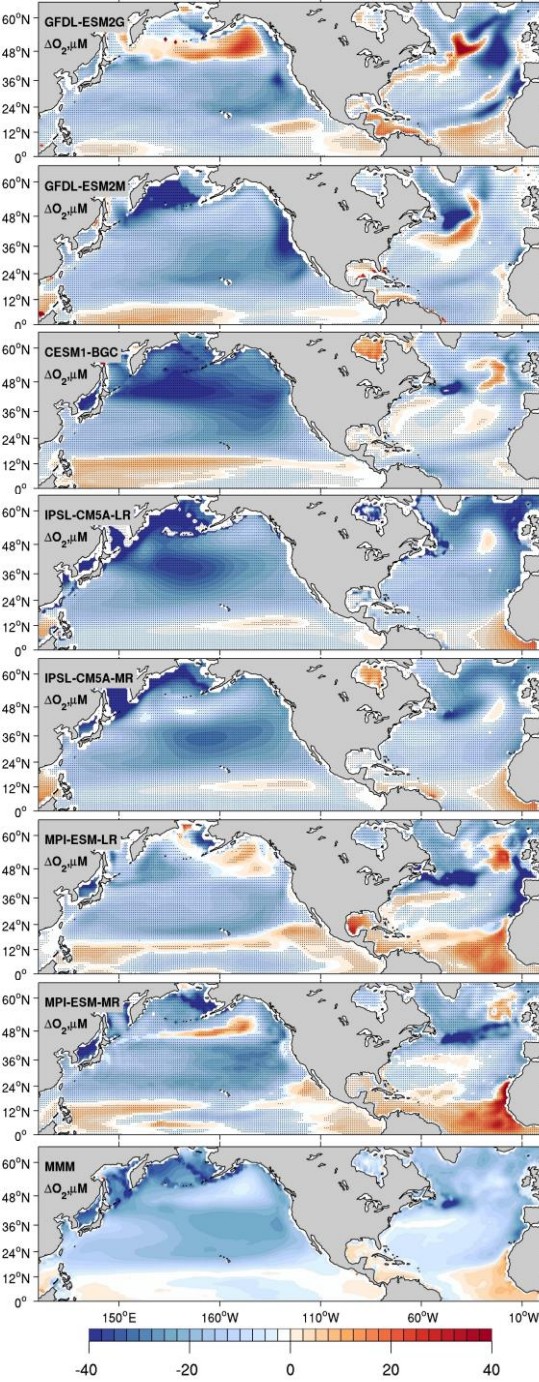

**Figure 5** Centennial change of dissolved oxygen calculated as the difference in 30-year averages between (2070-2100) and (1970-2000). All plotted values are 0-700 m depth averages. Drift is removed from the piControl simulation. Black dots indicate areas where the results are statistically significant at the 99% confidence level according to a t-test.

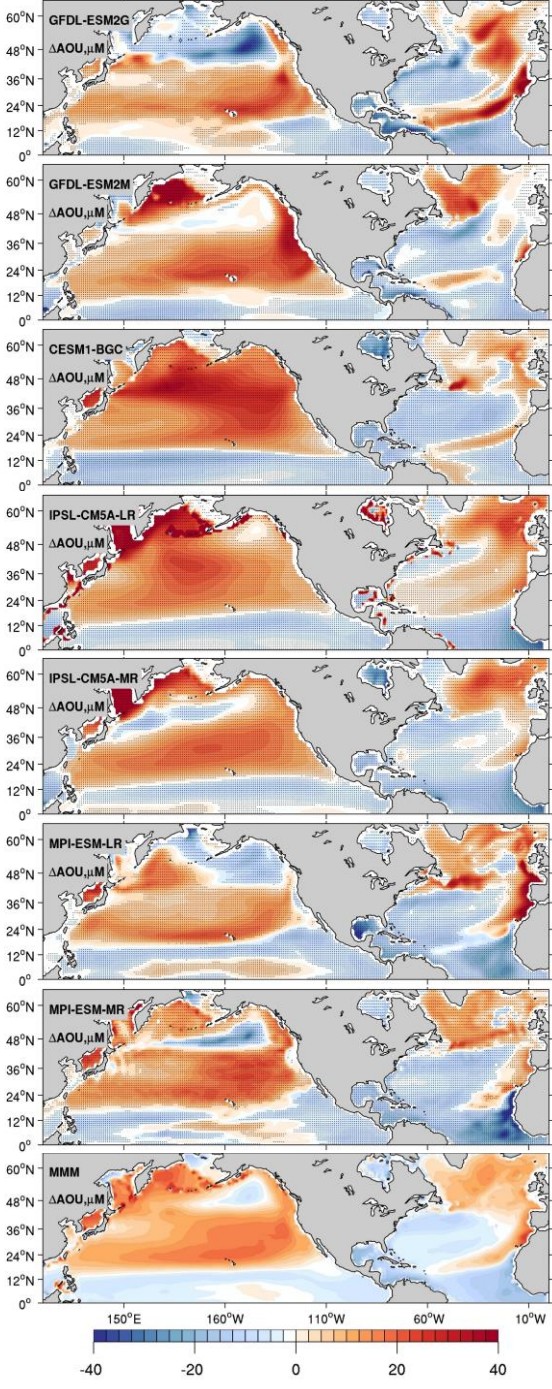

**Figure 6** Centennial change of apparent oxygen utilization calculated as the difference in 30-year averages between (2070-2100) and (1970-2000). All plotted values are 0-700 m depth averages. Drift is removed from the piControl simulation. Black dots indicate areas where the results are statistically significant at the 99% confidence level according to a t-test.

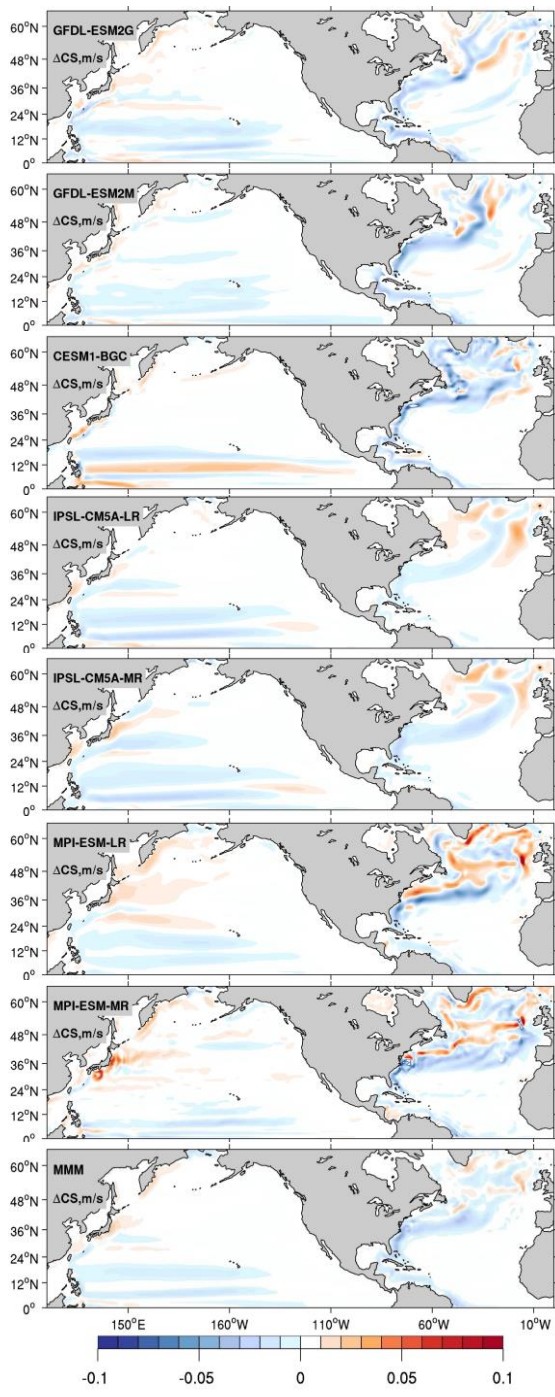

**Figure 7 Centennial change of current speed calculated as the difference in 30-year averages between (2070-2100) and (1970-2000). All plotted values are 0-700 m depth averages.**

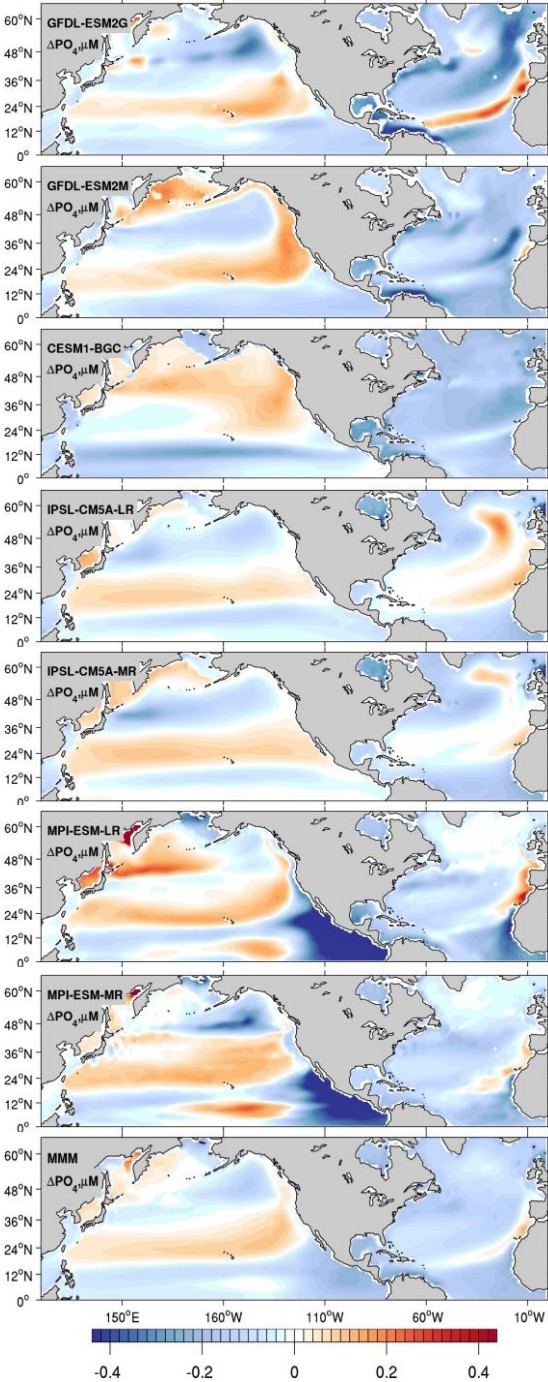

**Figure 8 Centennial change of PO₄ calculated as the difference in 30-year averages between (2070-2100) and (1970-2000). All plotted values are 0-700 m depth averages.**

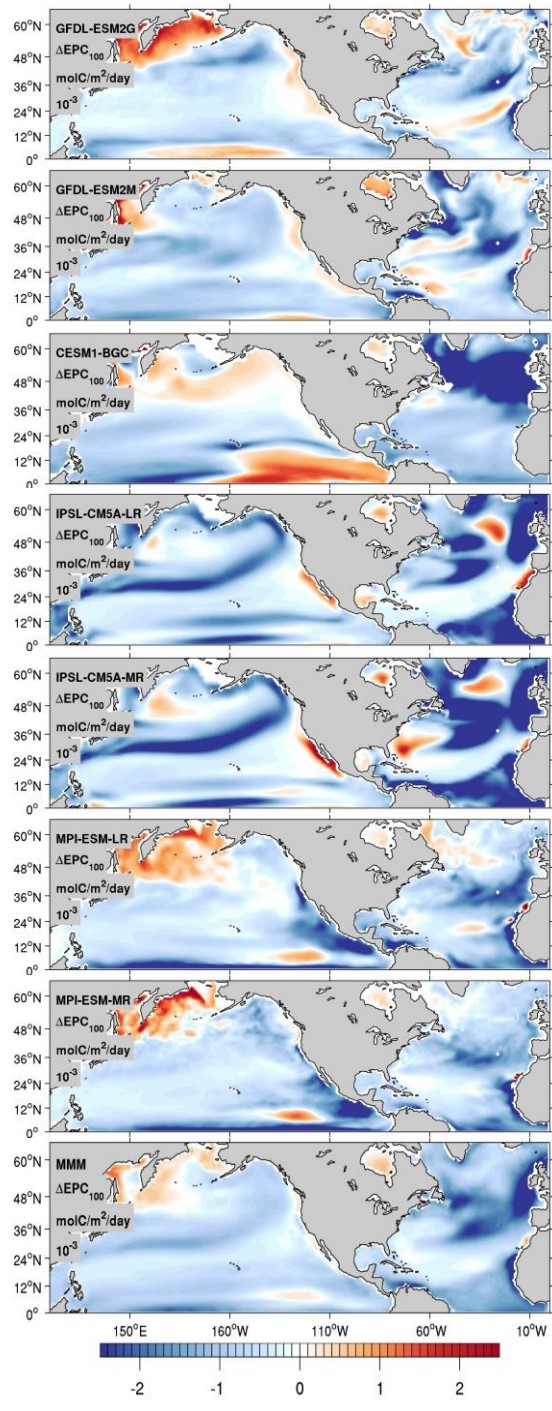

**Figure 9  Centennial change of export production calculated as the difference in 30-year averages between (2070-2100) and (1970-2000).**

**Table 1: Averaged changes of temperature ($\Delta T$), dissolved oxygen ($\Delta O_2$), oxygen solubility $\Delta(O_{2,sat})$, apparent oxygen utilisation $\Delta(AOU)$, and nutrient $\Delta(PO_4)$ between 10ºN-48ºN for Pacific and Atlantic basins averaged over the upper 0-700 m. The changes are calculated as the differences between the 30-year period 2070-2100 in the rcp8.5 scenario and 1970-2000 in the historical simulations.**

| | $\Delta T$ | | $\Delta O_2$ | | $\Delta O_{2,sat}$ | | $\Delta AOU$ | | $\Delta PO_4$ | |
| --- | --- | --- | --- | --- | --- | --- | --- | --- | --- | --- |
| | Pac | Atl | Pac | Atl | Pac | Atl | Pac | Atl | Pac | Atl |
| GFDL-ESM2G | 1.14 | 1.69 | -12.4 | -6.9 | -6.1 | -8.4 | 6.3 | -1.5 | -0.007 | -0.13 |
| GFDL-ESM2M | 1.14 | 2.21 | -16.1 | -6.1 | -5.8 | -11.0 | 10.1 | -5 | 0.001 | -0.15 |
| CESM1-BGC | 1.20 | 2.00 | -16.8 | -6 | -6.2 | -10.3 | 10.7 | -4.2 | -0.020 | -0.16 |
| IPSL-CM5A-LR | 1.60 | 2.03 | -18.1 | -12 | -8.6 | -10.0 | 9.5 | -1.7 | 0.000 | -0.006 |
| IPSL-CM5A-MR | 1.60 | 2.45 | -16.0 | -7 | -8.5 | -12.2 | 7.5 | -4.8 | -0.005 | -0.04 |
| MPI-ESM-LR | 1.07 | 1.90 | -10.0 | -6 | -5.1 | -9.5 | 4.6 | -3.6 | -0.001 | -0.10 |
| MPI-ESM-MR | 1.12 | 2.00 | -12.2 | -3.4 | -5.4 | -10.0 | 6.7 | -6.6 | 0.001 | -0.10 |
| MMM | 1.26 | 2.04 | -14.5 | -6.7 | -6.5 | -10.2 | 7.9 | -3.9 | -0.004 | -0.10 |

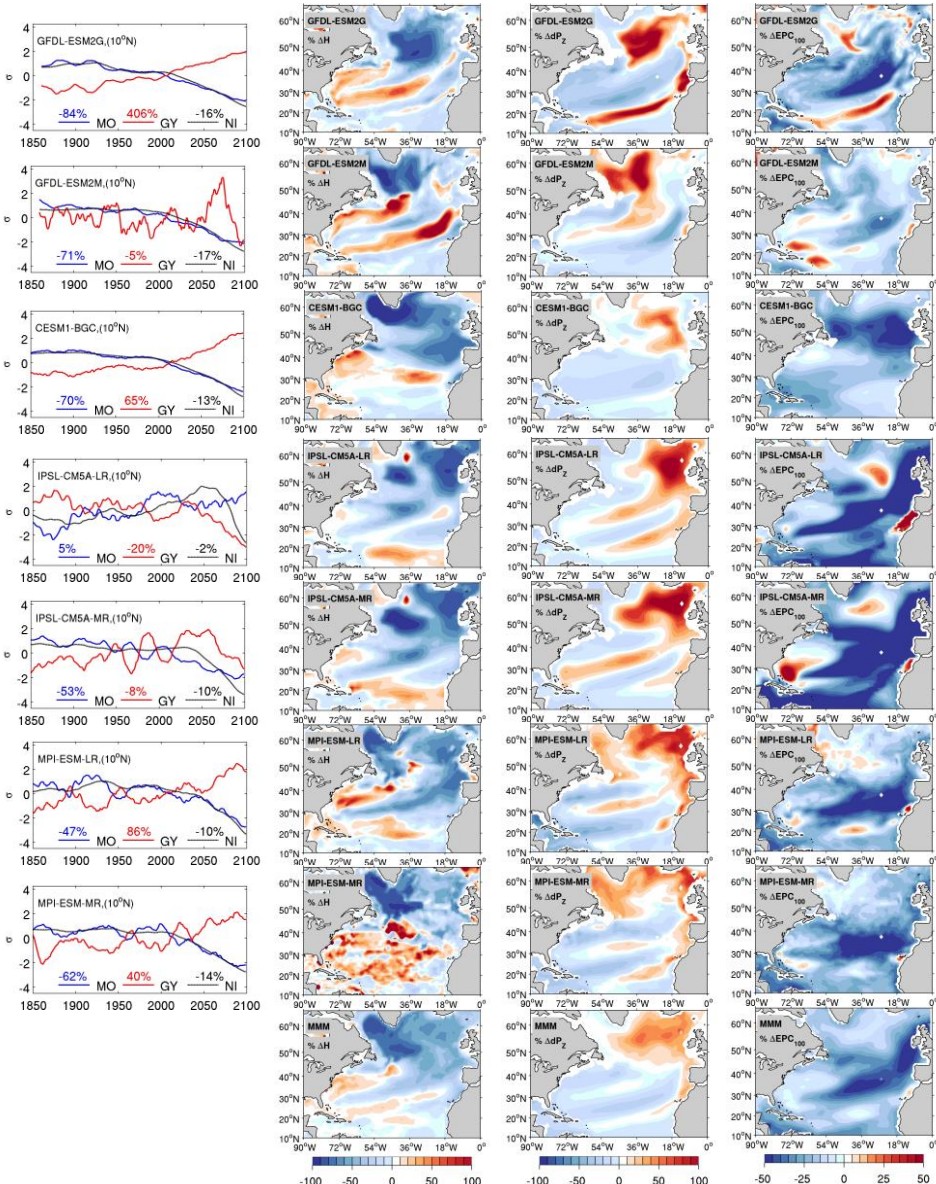

**Figure 10 (Left column) Normalized timeseries of zonally (70ºW-17ºW) and depth (0-700m) integrated northward nutrient transport components, ($MO = \overline{v}\,\overline{PO_4}$) and gyre ($GY = \overline{v'PO'_4}$), at 10ºN and nutrient inventory (NI) of the subtropical gyre (10ºN-48ºN, 0-700 meters). Coloured values represent the percent centennial change of each variable. Panels represent the percent centennial change of H (year maximum seasonal change of mixed layer depth), dPz (vertical gradient of $PO_4$), EPC$_{100}$ export production, all calculated as the difference in 30-year averages between (2070-2100) and (1970-2000).**

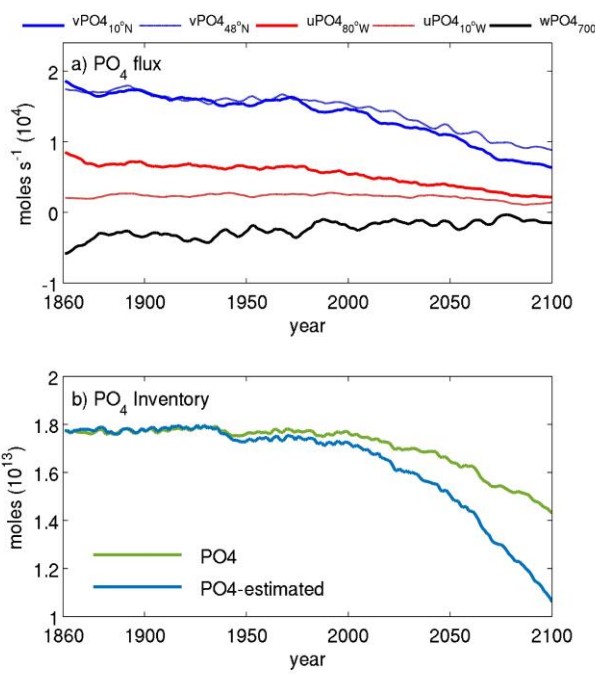

**Figure 11 a) Lateral nutrient fluxes in and out of a box enclosing the subtropical North Atlantic area with boundaries 10ºN-48ºN, 80ºW-10ºW and at 700 meters depth. All curved are in units of moles/s. b) Nutrient inventory estimated integrating lateral fluxes over time (light blue) and compared with the actual nutrient inventory (light green).**

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
