# Peer review of "Modulation of the North Atlantic Deoxygenation by The Slowdown of the Nutrient Stream"

_Biogeosciences, 2019_

## Referee Comment (RC1) · Anonymous Referee #1 · 10 Jun 2019

This study presents an analysis of output of dissolved O2, PO4, temperature, advective velocities, and sinking POC flux from four ESMs in the CMIP5 ensemble to investigate the drivers of upper ocean deoxygenation, comparing the North Atlantic to the North Pacific. The compensating effects of the temperature-controlled decrease in O2 solubility, ocean circulation/ventilation effects, and changes in biological O2 utilization are examined and attributed to identified trends in basin O2 content in the ESM outputs comparing the 1970-2000 period with the predicted 2070-2100 period under RCP 8.5 forcing. A contrasting pattern between the North Atlantic and North Pacific is identified, with deoxygenation proceeding more rapidly in the Pacific despite a smaller temperature increase in that basin. Solubility driven deoxygenation in the North Atlantic is revealed to be compensated for by a mechanism rooted in the slowdown of the AMOC,

which slows the important Gulf Stream nutrient stream, decreasing lateral injection of nutrients into the subtropical North Atlantic, decreasing export production and concomitant biological O2 utilization in the subsurface. Deoxygenation in the North Pacific is revealed to proceed via dual solubility and change in ventilation controls.

This is a well-written, clear, and concise manuscript detailing the authors study. I only have some comments regarding the Methods.

Line 81-83: What method was used to interpolate the model output to the common WOA grid?

Line 152-160: Why not perform a similar t-test as performed for O2 to test for statistical significance of the temperature increases identified?

Why were these 4 models chosen and not others from the CMIP5 ensemble? The given reason is that these 4 provided the variables of interest. Surely more than 4 CMIP5 models provide output of O2, PO4, temperature, advective velocities, and POC sinking flux for the historical and RCP8.5 cases?

Table 1: Why not include a column for the multi-model mean to be congruent with that which is provided in Figures 1-8?

Line 178: "The rate of solubility change...ranges from -12.07 $\mu$M to -14.81 $\mu$M" A rate implies a change with respect to another quantity or dimension, in this case time. I recommend switching these numbers to -0.12 $\mu$M/yr or mention the rate in parenthesis, etc.
* * *

---

## Referee Comment (RC2) · Anonymous Referee #2 · 14 Jun 2019

The study presents an analysis of phosphate, dissolved oxygen, temperature, apparent oxygen and velocity changes in the upper ocean for the northern hemisphere from 4 different CMIP5 Earth system model projections.

A plausible narrative is presented as to how changes in the dissolved oxygen in the northern basins are controlled in terms of overturning and temperature changes. There is a clear asymmetry between the North Atlantic and North Pacific with a weakening of the meridional overturning in the former basin. Taking that view forward, the narrative is engaging in terms of discussing the likely changes in terms of a weakening of the western boundary currents and the associated nutrient stream.

However, the study is lacking in providing supporting quantitative analysis to endorse the above interpretations as to how the dissolved oxygen and nutrient distributions are

controlled. There are no estimates of the nitrate flux carried by the western boundary currents and no estimates of nitrate transports along sections running across the basin. There is no real proof that either the nutrient transport change or the temperature changes are controlling the dissolved oxygen changes. Thus, a very plausible set of interpretations are presented that need to be made more quantitative and so become more robust and convincing.

I recommend that the authors address the following points: 1. What is the northward transport of nutrients in the nutrient stream and by how much has that weakened? 2. The nutrient stream is providing a redistribution of nutrients, but it is unclear whether the changes in this redistribution is confined within the subtropical gyre/subpolar gyre of the North Atlantic? 3. Alternatively, the changes in the overturning drives a weakening in the change in nutrient transport across the equator from the South to the North Atlantic.

For points 2 & 3, see Palter and Lozier (2008) supporting a more confined basin view and Sarmiento et al. (2004) and Williams et al. (2006) supporting a cross basin view. Both processes could be occurring to different extents.

While I agree with the view that the horizontal transport and redistribution of nutrients is probably key to the response, there needs to be mention of the implied changes in the vertical transfer of nutrients and oxygen within the basins.

A minor point, there are a lot of abbreviations that could be removed to make the text more readable.

In summary, the manuscript presents an engaging view of how the nutrients and dissolved oxygen distributions are controlled in the Earth system model projections. However, the authors need to provide more quantitative evidence to support their interpretations, rather than rely on changes in property maps.

References Palter, J. B., and M. S. Lozier (2008), On the source of Gulf Stream

nutrients, J. Geophys. Res., 113, C06018, doi:10.1029/2007JC004611. Sarmiento, J. L., N. Gruber, M. A. Brzezinski, and J. P. Dunne (2004), High‐latitude controls of thermocline nutrients and low latitude biological productivity, Nature, 427, 56–60. Williams, R. G., V. Roussenov, and M. J. Follows (2006), Nutrient streams and their induction into the mixed layer, Global Biogeochem. Cycles, 20, GB1016, doi:10.1029/2005GB002586.

---

## Author Comment (AC1) · 4 Jul 2019

**Nutrient Stream"**

by F. Tagklis, T. Ito, A. Bracco

*Texts in black are the original comments by the reviewers, which is followed by our responses in blue coloured text.

**Anonymous Referee #1:**

This study presents an analysis of output of dissolved O2, PO4, temperature, advective velocities, and sinking POC flux from four ESMs in the CMIP5 ensemble to investigate the drivers of upper ocean deoxygenation, comparing the North Atlantic to the North Pacific. The compensating effects of the temperature-controlled decrease in O2 solubility, ocean circulation/ventilation effects, and changes in biological O2 utilization are examined and attributed to identified trends in basin O2 content in the ESM outputs comparing the 1970-2000 period with the predicted 2070-2100 period under RCP 8.5 forcing. A contrasting pattern between the North Atlantic and North Pacific is identified, with deoxygenation proceeding more rapidly in the Pacific despite a smaller temperature increase in that basin. Solubility driven deoxygenation in the North Atlantic is revealed to be compensated for by a mechanism rooted in the slowdown of the AMOC, which slows the important Gulf Stream nutrient stream, decreasing lateral injection of nutrients into the subtropical North Atlantic, decreasing export production and concomitant biological O2 utilization in the subsurface. Deoxygenation in the North Pacific is revealed to proceed via dual solubility and change in ventilation controls. This is a well-written, clear, and concise manuscript detailing the authors study. I only have some comments regarding the Methods.

We appreciate the positive comments on our manuscript. We have addressed the questions related to the Methods below.

**Line 81-83: What method was used to interpolate the model output to the common WOA grid?**

The method used was bilinear interpolation using Climate Data Operators.

**Line 152-160: Why not perform a similar t-test as performed for O2 to test for statistical significance of the temperature increases identified?**

The significance of ocean warming under rcp8.5 in the CMIP5 has been already presented in previous works. For consistency in terms of analysis we would direct you to our previous study (Tagklis et al., 2017) and figures 1, 14 and 16. We think that a t-test for temperature is not necessary as repetitive, while

the significance of $O_2$ provides new information. We added a comment in the manuscript though where T plots are presented.

**Why were these 4 models chosen and not others from the CMIP5 ensemble? The given reason is that these 4 provided the variables of interest. Surely more than 4 CMIP5 models provide output of O2, PO4, temperature, advective velocities, and POC sinking flux for the historical and RCP8.5 cases?**

We could not find the complete set of variables/experiments for the rest of the models (EPC100 is often the culprit).

**Table 1: Why not include a column for the multi-model mean to be congruent with that which is provided in Figures 1-8?**

Following the reviewer's suggestion, we included the column to the table to be congruent with the rest of the presentation in the manuscript.

**Line 178: "The rate of solubility change ranges from -12.07 µM to -14.81 µM" A rate implies a change with respect to another quantity or dimension, in this case time. I recommend switching these numbers to -0.12 µM/yr or mention the rate in parenthesis, etc.**

We want to thank the reviewer for pointing this out. Indeed, the rate refers to a centennial time scale, which should be clarified. Furthermore, switching these numbers to an annual rate would imply a linear trend, which is not justified in the current analysis. We re-wrote the sentence as:

"The centennial solubility changes are calculated as differences between two 30 years-periods 1975-2005 and 2070-2100."

---

## Author Comment (AC2) · 4 Jul 2019

**Nutrient Stream"**

by F. Tagklis, T. Ito, A. Bracco

*Texts in black are the original comments by the reviewers, which is followed by our responses in blue coloured text.

**Anonymous Referee #2:**

The study presents an analysis of phosphate, dissolved oxygen, temperature, apparent oxygen and velocity changes in the upper ocean for the northern hemisphere from 4 different CMIP5 Earth system model projections. A plausible narrative is presented as to how changes in the dissolved oxygen in the northern basins are controlled in terms of overturning and temperature changes. There is a clear asymmetry between the North Atlantic and North Pacific with a weakening of the meridional overturning in the former basin. Taking that view forward, the narrative is engaging in terms of discussing the likely changes in terms of a weakening of the western boundary currents and the associated nutrient stream. However, the study is lacking in providing supporting quantitative analysis to endorse the above interpretations as to how the dissolved oxygen and nutrient distributions are controlled. There are no estimates of the nitrate flux carried by the western boundary currents and no estimates of nitrate transports along sections running across the basin. There is no real proof that either the nutrient transport change or the temperature changes are controlling the dissolved oxygen changes. Thus, a very plausible set of interpretations are presented that need to be made more quantitative and so become more robust and convincing.

I recommend that the authors address the following points:

**1. What is the northward transport of nutrients in the nutrient stream and by how much has that weakened?**

**2. The nutrient stream is providing a redistribution of nutrients, but it is unclear whether the changes in this redistribution is confined within the subtropical gyre/subpolar gyre of the North Atlantic?**

**3. Alternatively, the changes in the overturning drives a weakening in the change in nutrient transport across the equator from the South to the North Atlantic.**

For points 2 & 3, see Palter and Lozier (2008) supporting a more confined basin view and Sarmiento et al. (2004) and Williams et al. (2006) supporting a cross basin view. Both processes could be occurring to different extents.

While I agree with the view that the horizontal transport and redistribution of nutrients is probably key to the response, there needs to be mention of the implied changes in the vertical transfer of nutrients and oxygen within the basins.

**A minor point, there are a lot of abbreviations that could be removed to make the text more readable.**

In summary, the manuscript presents an engaging view of how the nutrients and dissolved oxygen distributions are controlled in the Earth system model projections. However, the authors need to provide more quantitative evidence to support their interpretations, rather than rely on changes in property maps.

We appreciate reviewer's comments. We have tried to address and clarify the suggested points as indicated below. In doing so, we had to consider key limitations in the variables that are available in the CMIP5 catalogue and the frequency at which they were saved.

Regarding points 1, 2, and 3:

We discuss in the introduction of the manuscript the role of the nutrient streams in biogeochemical cycling and their  contribution towards maintaining basin scale productivity at the mid- and high-latitudes over interannual and longer timescales (Letscher et al., 2016; Palter and Lozier, 2008; Palter et al., 2005; Williams and Follows, 1998; Williams et al., 2011; Williams et al., 2006).

Focusing only on centennial-scale changes, we observed a common pattern among the models suggesting a plausible mechanism of the ocean resisting to the deoxygenation as a result of changes in the large-scale ocean circulation. As indicated by the reviewer, the study is lacking a more quantitative analysis of the fluxes and advective transport. The calculation of advective fluxes and budget analysis require monthly model output. Unfortunately, nutrient and oxygen data are typically saved at such frequency only at the surface in the CMIP5 catalogue and as annual averages at all other depths. So it is impossible to perform these analyses realiably for this manuscript. We added a sentence in this regard in the Conclusions of the paper.

Attempting to address the reviewer's comment on the nutrient transport changes, we looked for an additional quantitative analyses in our assessment.  In figure S1 we constructed an index of phosphate concentration in the upper 0-700m in a box area covering the extent of the western boundary currents. We then calculated the correlation coefficient of such index with the apparent oxygen utilization, also averaged over the 0-700m layer. The boxes used to construct the indices are different for the two basins but the same among the models. The time series extend over the historical period 1870-2005. The correlation map

provides supportive evidence of the linearly correlated behavior between the western boundary currents, phosphate (nutrients) and AOU concentrations.

**Further supporting evidence is provided in figure S3 where we present the regression coefficients of the current speed for the period (1870-2005) onto the upper ocean (0-700m) phosphate index defined in the box indicated in each basin (box PO4). The phosphate index lags 2-years the current speed timeseries.

To prove the role of temperature on oxygen changes we could provide the centennial changes of the O2 saturation which is inversely proportional to the temperature changes, but we decided to exclude this figure as repetitive. Indeed it does not provide more information than the $\Delta T$ maps as shown below in Figures S5,S6.

[Figure]

Figure S1: Regression coefficients of the apparent oxygen utilization (AOU), for the period (1870-2005) esmHistorical annual mean output onto the upper ocean (0-700m) phosphate index defined in the box indicated in each basin (box PO4). The contour represents 0-700 meters and 1987-2005 averaged current speed greater than 0.05 m/s and reflects the Western Boundary and North Atlantic Current pathway.

Atlantic Ocean Box:    Lon: [69  49] West,    Lat: [36.5  43.5] North

Pacific Ocean Box:    Lon: [135  155] East,    Lat: [31.5  38.5] North

[Figure]

Figure S3: (As in figure S1)Regression coefficients of the apparent oxygen utilization (AOU), for the period (1870-2005) esmHistorical annual mean output onto the upper ocean (0-700m) phosphate index defined in the box indicated in each basin (box PO4). The contour represents 0-700 meters and 1987-2005 averaged current speed greater than 0.05 m/s and reflects the Western Boundary and North Atlantic Current pathway.

[Figure]

Figure S4: Regression coefficients of the current speed for the period (1870-2005) esmHistorical annual mean output onto the upper ocean (0-700m) phosphate index defined in the box indicated in each basin (box PO4). **The phosphate index lags 2-years the CS timeseries.** The contour represents 0-700 meters and 1987-2005 averaged current speed greater than 0.05 m/s and reflects the Western Boundary and North Atlantic Current pathway.

[Figure]

Figure S5: Centennial change of oxygen saturation calculated as the difference in 30-year averages between (2070-2100) and (1970-2000). All plotted values are 0-700 m depth averages.

[Figure]

Figure S6: Centennial change of T calculated as the difference in 30-year averages between (2070-2100) and (1970-2000). All plotted values are 0-700 m depth averages.

**References**

Letscher, R.T., Primeau, F., Moore, J.K. (2016) Nutrient budgets in the subtropical ocean gyres dominated by lateral transport. Nature Geoscience 9, 815.

Palter, J.B., Lozier, M.S. (2008) On the source of Gulf Stream nutrients. Journal of Geophysical Research-Oceans 113.

Palter, J.B., Lozier, M.S., Barber, R.T. (2005) The effect of advection on the nutrient reservoir in the North Atlantic subtropical gyre. Nature 437, 687.

Tagklis, F., Bracco, A., Ito, T. (2017) Physically driven patchy O2 changes in the North Atlantic Ocean simulated by the CMIP5 Earth system models. Global Biogeochemical Cycles 31, 1218-1235.

Williams, R.G., Follows, M.J. (1998) The Ekman transfer of nutrients and maintenance of new production over the North Atlantic. Deep-Sea Research Part I-Oceanographic Research Papers 45, 461-489.

Williams, R.G., McDonagh, E., Roussenov, V.M., Torres-Valdes, S., King, B., Sanders, R., Hansell, D.A. (2011) Nutrient streams in the North Atlantic: Advective pathways of inorganic and dissolved organic nutrients. Global Biogeochemical Cycles 25.

Williams, R.G., Roussenov, V., Follows, M.J. (2006) Nutrient streams and their induction into the mixed layer. Global Biogeochemical Cycles 20.

---

## Author Response (AR1)

**"Modulation of the North Atlantic Deoxygenation by The Slowdown of the**

**Nutrient Stream"**

by F. Tagklis, T. Ito, A. Bracco

*In black the original comments by the editor, followed by our responses in blue.

**Associate Editor Decision: Reconsider after major revisions** (17 Jul 2019) by Andreas Oschlies**:**

Comments to the Author:

The manuscript is a nice overview over northern hemisphere upper ocean deoxygenation as simulated by a few CMIP5 models. It concentrates on the different declines in the upper North Atlantic (small) and North Pacific (large). The slowdown of the North Atlantic nutrient stream is suggested to explain the different deoxygenation rates in the two basins. This provides an interesting story, which I find plausible and worthwhile of eventual publication. However, all arguments are based on relatively vague qualitative descriptions of basin scale patterns of simulated changes rather than being mechanistically 'confirmed' (line 199) or quantiatively estimated as providing 'most' of the simulated deoxygenation (line 222). There is no mentioning of the possible role of other processes (e.g. changes in up- or downwelling, subpolar gyre strength, fresh water supply and stratification, carbon export by vertical mixing (including deep convection in the N.Atlantic) of particulate or organic matter - are these included in the diagnosed export production of Fig. 9?).

Two expert reviewers have provided useful comments and thoughtful criticism, which I find valid. Unfortunately, the current responses of the authors fail short of satisfactorily addressing those points. Saying that the required model fields are unfortunately not available is not an appropriate scientific reasoning (it would only say that the study was ill-designed - which, I think, is not the case). One could, for example, use one model for which the required temporal resolution output is available and estimate the errors made when considering annual mean tracer fields only. I therefore would like to ask the authors to carefully address all the points raised by the reviewers and also discuss possible other mechanisms as well. I expect that this will become a scientifically very useful contribution and look forward to seeing a revised version.

Many thanks and best regards, -Andreas

**Response to Editor's Comments**

We much appreciate your time assessing the manuscript and the review comments. Most of the issues raised by the two expert reviewers were addressed in the revised manuscript and the following response letter.

We want to briefly outline the main changes:

- Three more models were added in the analysis.
- Figures and table were updated accordingly
- Table 1: We changed the northern boundary from 50ºN to 48ºN, thus some minor changes in the numbers.
- We provide additional analysis with a more quantitative character on the mechanisms in the last section of the manuscript, leading to the likely causes of the reduction in AOU of the subtropical North Atlantic. More details are given in the Response to Rev. 2.
- We provide additional analysis and discussion on the possible role of other processes in controlling the AOU of the Atlantic Basin. More details are provided in the Response to Rev. 2

Please see below a more detailed response to reviewer's comments.

Best regards,

On behalf of co-authors,

Filippos Tagklis

**Interactive comment on:**

**"Modulation of the North Atlantic Deoxygenation by The Slowdown of the Nutrient Stream"**

by F. Tagklis, T. Ito, A. Bracco

*In black are the original comments by the reviewers, which is followed by our responses in blue.

**Anonymous Referee #1:**

This study presents an analysis of output of dissolved O2, PO4, temperature, advective velocities, and sinking POC flux from four ESMs in the CMIP5 ensemble to investigate the drivers of upper ocean deoxygenation, comparing the North Atlantic to the North Pacific. The compensating effects of the temperature-controlled decrease in O2 solubility, ocean circulation/ventilation effects, and changes in biological O2 utilization are examined and attributed to identified trends in basin O2 content in the ESM outputs comparing the 1970-2000 period with the predicted 2070-2100 period under RCP 8.5 forcing. A contrasting pattern between the North Atlantic and North Pacific is identified, with deoxygenation proceeding more rapidly in the Pacific despite a smaller temperature increase in that basin. Solubility driven deoxygenation in the North Atlantic is revealed to be compensated for by a mechanism rooted in the slowdown of the AMOC, which slows the important Gulf Stream nutrient stream, decreasing lateral injection of nutrients into the subtropical North Atlantic, decreasing export production and concomitant biological O2 utilization in the subsurface. Deoxygenation in the North Pacific is revealed to proceed via dual solubility and change in ventilation controls. This is a well-written, clear, and concise manuscript detailing the authors study. I only have some comments regarding the Methods.

We appreciate the positive comments on this manuscript. We have addressed your questions related to the Methods below.

**Line 81-83: What method was used to interpolate the model output to the common WOA grid?**

The method used was bilinear interpolation using Climate Data Operators. This information has been added to the text.

**Line 152-160: Why not perform a similar t-test as performed for O2 to test for statistical significance of the temperature increases identified?**

Following reviewer's suggestion, we performed a similar t-test for temperature and apparent oxygen utilization in figures 4 and 6 in the main text.

**Why were these 4 models chosen and not others from the CMIP5 ensemble? The given reason is that these 4 provided the variables of interest. Surely more than 4 CMIP5 models provide output of O2, PO4, temperature, advective velocities, and POC sinking flux for the historical and RCP8.5 cases?**

We added three more models in the analysis. Now, the subset consists of 7 models in total. We looked into two more models, IPSL-CM5B-LR which appears to have significant errors, and HADGEM-CC which during the evaluation appeared to have very low values of nutrient in the Atlantic basin, but most importantly the monthly maximum ocean mixed layer depth variable 'omlmax' was not available in any of the ESGF-CoG nodes. This variable was used in the additional analysis. We decided to exclude both.

**Table 1: Why not include a column for the multi-model mean to be congruent with that which is provided in Figures 1-8?**

Following the reviewer's suggestion, we included the suggested column to the table to be congruent with the rest of the presentation in the manuscript. The table has been updated with the additional models.

**Line 178: "The rate of solubility change ranges from -12.07 μM to -14.81 μM" A rate implies a change with respect to another quantity or dimension, in this case time. I recommend switching these numbers to -0.12 μM/yr or mention the rate in parenthesis, etc.**

We want to thank the reviewer for pointing this out. Indeed, the rate refers to a centennial time scale. We clarified in the revision. Switching these numbers to an annual rate would imply a linear trend, which is not justified in the current analysis. We mention in all figures and table that changes are calculated as differences between two 30 years-periods 1970-2000 and 2070-2100.

**Interactive comment on:**

**"Modulation of the North Atlantic Deoxygenation by The Slowdown of the Nutrient Stream"**

by F. Tagklis, T. Ito, A. Bracco

*In black are the original comments by the reviewers, followed by our responses in blue.

**Anonymous Referee #2:**

The study presents an analysis of phosphate, dissolved oxygen, temperature, apparent oxygen and velocity changes in the upper ocean for the northern hemisphere from 4 different CMIP5 Earth system model projections. A plausible narrative is presented as to how changes in the dissolved oxygen in the northern basins are controlled in terms of overturning and temperature changes. There is a clear asymmetry between the North Atlantic and North Pacific with a weakening of the meridional overturning in the former basin. Taking that view forward, the narrative is engaging in terms of discussing the likely changes in terms of a weakening of the western boundary currents and the associated nutrient stream. However, the study is lacking in providing supporting quantitative analysis to endorse the above interpretations as to how the dissolved oxygen and nutrient distributions are controlled. There are no estimates of the nitrate flux carried by the western boundary currents and no estimates of nitrate transports along sections running across the basin. There is no real proof that either the nutrient transport change or the temperature changes are controlling the dissolved oxygen changes. Thus, a very plausible set of interpretations are presented that need to be made more quantitative and so become more robust and convincing.

I recommend that the authors address the following points:

**1. What is the northward transport of nutrients in the nutrient stream and by how much has that weakened?**

**2. The nutrient stream is providing a redistribution of nutrients, but it is unclear whether the changes in this redistribution is confined within the subtropical gyre/subpolar gyre of the North Atlantic?**

**3. Alternatively, the changes in the overturning drives a weakening in the change in nutrient transport across the equator from the South to the North Atlantic.**

For points 2 & 3, see Palter and Lozier (2008) supporting a more confined basin view and Sarmiento et al. (2004) and Williams et al. (2006) supporting a cross basin view. Both processes could be occurring to different extents.

While I agree with the view that the horizontal transport and redistribution of nutrients is probably key to the response, there needs to be mention of the implied changes in the vertical transfer of nutrients and oxygen within the basins.

**A minor point, there are a lot of abbreviations that could be removed to make the text more readable.**

In summary, the manuscript presents an engaging view of how the nutrients and dissolved oxygen distributions are controlled in the Earth system model projections. However, the authors need to provide more quantitative evidence to support their interpretations, rather than rely on changes in property maps.

We appreciate reviewer's comments. We have tried to address and clarify the suggested points as follows.

Regarding points 1, 2, and 3:

We estimated changes of the northward supply of phosphate at 10ºN along with nutrient inventory of the subtropical gyre. In the new Figure 10 of the revised manuscript, time series represent the zonally and vertically integrated northward transport of phosphate $(\overline{vPO_4})$ at 10ºN over the 0-700 meters depth range, decomposed in the overturning $(MO = \bar{v}\overline{PO_4})$ and gyre $(GY = \overline{v'PO'_4})$ components, along with the nutrient inventory (NI) zonally, meridionally and depth-integrated over 10ºN-48ºN and 0-700 meters. The overbar indicates the zonal mean, and the primes indicate the departure from the zonal mean. For better comparison, we applied a low pass filter of 10-years, and then we normalised the time series by subtracting their mean and dividing by their standard deviation. The coloured values [Figure 10] represent the per cent centennial change of each transport component and nutrient inventory. The subtropical gyre nutrient inventory closely follows the declining trajectory of the overturning component of the northward nutrient transport at 10ºN for all models but IPSL-CM5A-LR.

Regarding the vertical transport of nutrients in the euphotic layer, we additionally explored the changes in the vertical entrainment of thermocline nutrients. Results suggest that different process are at play in the subpolar versus subtropical the gyre, which we describe in the manuscript (line 243, Figure 10).

Regarding the comment "There is no real proof that either the nutrient transport change or the temperature changes are controlling the dissolved oxygen changes.". To convince the reviewer about the role of temperature on oxygen changes we provide here the centennial changes of the O2 saturation $(\Delta O_{2,sol})$ which is inversely proportional to the temperature changes, but we decided to exclude this figure as repetitive. Indeed, it does not provide more information than the $\Delta T$ maps as shown below in figure S1 to be compared to figure 4 in the main text.

[revised manuscript text omitted]

---

## Author Response (AR2)

**Nutrient Stream"**

by F. Tagklis, T. Ito, A. Bracco

*In black are the original comments by the reviewers, which is followed by our responses in blue.

**Anonymous Referee #1: (Major Revision #2)**

Line 84: Please provide a citation for Climate Data Operators.

Figure 5: I do not see the black dots indicating significantly regions as described in the caption.

Line 181: Suggested edit: 'temperature change by 1ºC' to 'temperature increase by 1ºC'. A change can be in either direction, but the statement concludes with an O2 solubility decrease, implying a temperature increase.

We have changed and clarified the suggested points in lines 84, Figure 5 and line181 accordingly. The size of the dots representing statistically significant changes was also changed for figures 4, 5, 6.

**Interactive comment on:**

**"Modulation of the North Atlantic Deoxygenation by The Slowdown of the**
**Nutrient Stream"**

by F. Tagklis, T. Ito, A. Bracco

*In black are the original comments by the reviewers, which is followed by our responses in blue.

**Anonymous Referee #2:  (Major Revision #2)**

The study examines the centennial changes in upper ocean phosphate and export production from a suite of 7 CMIP5 models. There is a centennial decline in the upper ocean phosphate in all the models apart from the IPSL ones for the North Atlantic. This change is qualitatively associated with a weakening in the meridional ocean overturning and its nutrient stream. This connection is plausible, but is not fully confirmed in this study.

The authors have strengthened the manuscript by including 3 additional models and an analysis of the northward transport of nitrate at 10N over the upper 700m for all the models (in Fig. 10). The inclusion of the nitrate transport shows a decline in the normalised northward component associated with the meridional overturning in all the models apart from the IPSL-CM5A-LR one for the North Atlantic. This signal again lends support to the authors' hypothesis that the decline in the nutrient stream leads to the decline in nutrients in the North Atlantic basin.

However, as raised by the reviewers and the Editor, there are a range of other processes that might also be associated with the nutrient decline. I suspect that the authors are right in terms of their interpretation, but a higher level of evidence needs to be reached to support this plausible assertion.

In my view, a further simple calculation is required where the convergence of the Atlantic nitrate transport between 10N and a northern latitude circle is compared with the tendency in the mass-weighted nutrients within that enclosed volume, such that both terms have the same units (like mol/s). While there need not be an exact match due to other contributions, to be fully convincing, this comparison should at least have both the terms having a comparable magnitude and sign.

The authors may think that they have already shown that comparison, but the side panels in Fig. 10 only show the normalised time series of the nitrate transport and the nitrate inventory, which is not as convincing as the actual contributions to the budget, where terms have the same units and a convergence

in nitrate transport should equate to a positive tendency.

Including this budget comparison (without the normalisation) would remove doubts as whether the agreement in the sign in the decline in the nutrient stream and the nutrient stock is significant. The work would be more convincing if applied to all the models and I would have thought would be relatively straight forward given the rest of the work that has been completed; however, alternatively, I would be happy to see a more detailed nutrient budget performed for one of the representative models.

I recommend that the authors take on board this recommendation so that they are then able to provide a more convincing and substantial contribution, which is identifying an important climate change mechanism affecting biological productivity.

We appreciate the reviewer's comments. We have tried to address and clarify the suggested points as follows.

Following the reviewer's suggestion, we include an estimate of the regional nutrient budget using GFDL-ESM2M as a representative model. Following the geographical boundaries of the previous analysis, we consider a control volume enclosing the STNA with the boundaries at approximately $10^o$N-$48^o$N, $80^o$W-$10^o$W and 700 meters depth using the native model grid. We calculate and present all lateral nutrient transport terms in and out of the control volume, all in units of moles per second. Zonal and meridional fluxes are defined positive eastward and northward, and the vertical flux is defined positive upward.

Figure 11a shows the changes in magnitude and the sign of each component during the historical period 1861-2005 and rcp8.5 period 2006-2100. The northward supply of nutrients at $10^o$N ($vPO4_{(10oN)}$) has the most significant decline among the lateral transport terms and a comparable magnitude with the northward transport of nutrient at $48^o$N ($vPO4_{(48oN)}$). The western boundary transport component at $80^o$W ($uPO4_{(80^oW)}$) represents the net nutrient supply through the Florida current which loses half of its magnitude by the end of the $21^{st}$ century. The eastern boundary component at $10^o$W ($uPO4_{(10^oW)}$) remains mostly unchanged. The vertical component at 700 meters depth ($wPO4_{(700m)}$) is negative (downwelling) with decreasing magnitudes. The signs and the magnitude of the changes in the lateral and vertical transport terms are consistent with the weakened advective nutrient transport, providing additional support to our hypotheses.

The flux convergence of the (resolved) advective transport must be balanced by the time derivative of the nutrient inventory and the net biological nutrient sources/sinks. Sub-grid scale parameterizations could also contribute to the regional nutrient budget. It is difficult to precisely close the nutrient budget with the

available dataset. However, we can still integrate over time the flux convergence of the advective transport to calculate the 'estimated' nutrient inventory (PO4$_{estimated}$). Net advective convergence is positive, and its integral gradually increases over time because it does not include the nutrient uptake and export by biological processes (Bio). To account for the baseline pre-industrial biological component (Bio), we first determine the residual between the PO4$_{estimated}$ and the PO4 as Residual (=PO4$_{estimated}$ - PO4). Then the pre-industrial estimate of Bio is estimated as a linear trend based on the first 60 years (1861-1920) of the Residual. Then the corrected PO4$_{estimated}$ is determined as the temporal integral of the advective flux convergence minus Bio. Then the PO4$_{estimated}$ time series reflects the change in STNA nutrient budget if there were no changes in biological sources/sinks (constant Bio). Figure 11b shows that the decline of PO4$_{estimated}$ is much larger than that of PO4. After the year 2005 and during the rcp8.5 period the actual PO4 inventory does not decrease as much as the estimated inventory PO4$_{estimated}$ due to the weakened biological export of nutrients. This result is consistent with the weakened biological productivity of the North Atlantic as well as the circulation change as the main driver of the nutrient decline in the basin.

$$\frac{d(PO4)}{dt} = vPO4_{(10^oN)} - vPO4_{(48^oN)} + uPO4_{(80^oW)} - uPO4_{(10^oW)} + wPO4_{(700\ m)} - Bio \quad (eq.1)$$

[Figure]

**Figure 11 a) Lateral nutrient fluxes in and out of a box enclosing the subtropical North Atlantic area with boundaries 10ºN-48ºN, 80ºW-10ºW and 700 meters depth. All in units of moles/sec. b) Nutrient inventory integrating lateral fluxes over time (light blue) and compared with the actual nutrient inventory (light green).**